# Genome-Wide Survey and Analysis of Microsatellites in Waterlily, and Potential for Polymorphic Marker Development

**DOI:** 10.3390/genes13101782

**Published:** 2022-10-02

**Authors:** Xiang Huang, Meihua Yang, Jiaxing Guo, Jiachen Liu, Guangming Chu, Yingchun Xu

**Affiliations:** 1College of Agriculture, Shihezi University, Shihezi 832003, China; 2College of Horticulture, Nanjing Agricultural University, Nanjing 210095, China

**Keywords:** *Nymphaea colorata*, genome-wide mining, microsatellites, polymorphism markers

## Abstract

Waterlily (Nymphaeaceae), a diploid dicotyledon, is an ornamental aquatic plant. In 2020, the complete draft genome for the blue-petal waterlily (*Nymphaea colorata*) was made available in GenBank. To date, the genome-wide mining of microsatellites or simple sequence repeats (SSRs) in waterlily is still absent. In the present study, we investigated the characteristics of genome-wide microsatellites for *N. colorata* and developed polymorphic SSR markers across tropical and hardy waterlilies. A total of 238,816 SSRs were identified in 14 *N. colorata* chromosomes with an average density of 662.60 SSRs per Mb, and the largest number of SSRs were present on chromosome 1 (n = 30,426, 705.94 SSRs per Mb). The dinucleotide was the most common type, and AT-rich repeats prevail in the *N. colorata* genome. The SSR occurrence frequencies decreased as the number of motif repeats increased. Among 2442 protein-coding region SSRs, trinucleotides, accounting for 63.84%, were the most abundant. Gene ontology terms for signal transduction (e.g., GO: 0045859 and GO: 0019887) and the lipoic acid metabolism (ko00785,) were overrepresented in GO and KEGG enrichment analysis, respectively. In addition, 107,152 primer pairs were identified, and 13 novel polymorphism SSR markers were employed to distinguish among nine waterlily cultivars, of which Ny-5.2 and Ny-10.1 were the most informative SSR loci. This study contributes the first detailed characterization of SSRs in *N. colorata* genomes and delivers 13 novel polymorphism markers, which are useful for the molecular breeding strategies, genetic diversity and population structure analysis of waterlily.

## 1. Introduction

Waterlily, belonging to the Nymphaeaceae family, is an ornamental aquatic plant. There are at least 50 species in this genus and more than 1000 horticultural cultivars, with a worldwide distribution in tropical to temperate regions [1,2]. According to its eco-physiological characteristics, it can be divided into tropical and hardy waterlily [3]. Among these, blue-petal waterlily (*Nymphaea colorata*) is a typical tropical waterlily possessing gorgeous flowers and a high ornamental value [4]. Meanwhile, *Nymphaea candida*, a species native to Xinjiang Province (northwestern China), is a typically hardy waterlily with strong cold resistance and high biological significance [3].

In previous studies, various methods of molecular markers, such as amplified fragment length polymorphism (AFLP), inter-simple sequence repeat (ISSR), random amplified polymorphic DNA (RAPD) and simple sequence repeat (SSR), have been applied to the study of aquatic plant genomics, such as *Nelumbo*
*nucifera*, *Euryale ferox* and *Ranunculus nipponicus* [5,6,7]. Among them, SSR markers have the advantages of high polymorphism, genome richness and co-dominance in genetic diversity and phylogenetic analysis. Meanwhile, genome-wide microsatellite mining has been carried out on many species, such as *Brassica napus* [8], *Punica granatum* [9], *Brassica oleracea* [10], *Fagopyrum tataricum* [11] and *Anemone coronaria* [12], which offered novel molecular markers for their genetic improvement and genetic diversity studies. Moreover, in aquatic plants, genome-wide SSR markers of *N. nucifera* [6] were mined and polymorphic primers were successfully developed to analyze the inner-species difference and genetic similarities between cultivated and wild lotus.

*N. colorata*, a diploid dicotyledon, lies at the base of the angiosperm linage and has a strong genetic significance. At present, there are many varieties of waterlily species and commercial species, but there are few studies on the specific molecular markers of *Nymphaea* [13,14,15] or the molecular-level identification of different varieties. Therefore, it is crucial to analyze the characteristics of the simple repeat sequence (SSR) loci of the whole-genome sequence of *N. colorata* to identify germplasm resources, analyze genetic diversity and construct a genetic linkage map of *Nymphaea*. In 2020, the *N. colorata* genome was discovered, which has a relatively small size of 409.472 Mb [16]. This finding, a research milestone, provides a chance to develop research on molecular markers, comparative genomics and functional genomics in waterlily.

In this study, we aimed to investigate genome-wide SSRs in *N. colorata* by an analysis of SSR density, occurrence frequency and the possible functions of SSRs in CDS regions. Furthermore, polymorphic SSR markers were developed to distinguish multiple waterlily cultivars. Our study provides a useful tool for molecular breeding strategies, genetic diversity and the population structure analysis of waterlily.

## 2. Materials and Methods

### 2.1. Source of Genomic Sequences

*N. colorata* genome assemble sequences were obtained from the National Center for Biotechnology Information (NCBI), and its accession number is GCF_008831285.1. In addition, the annotation file (7.6 Mb) of the *N. colorata* genome was downloaded from https://ftp.ncbi.nlm.nih.gov/genomes, accessed on 15 October 2020.

### 2.2. Identification of Microsatellite and Primer Design

The microsatellite identification software MISA (http://pgrc.ipk-gatersleben.de/misa, accessed on 20 November 2020) was used to identify genome-wide and coding region microsatellites in the *N. colorata* genome sequences. To identify perfect microsatellites, the minimum repeat number was defined as 10 for mono-, 6 for di- and 5 for tri-, tetra-, penta- and hexa-nucleotide SSRs [17,18]. The flanking sequences of SSRs were used as targets for the primer design by the Primer3 program (http://frodo.wi.mit.edu/, accessed on 6 May 2021) with a MISA-generated Primer3 input file [19], and the major selecting parameters for primer design were as follows: primer length, 18-23 bp, with 20 bp being optimal; PCR product size, 100–500 bp, with 200 bp being optimal; an annealing temperature of 50–65 °C; and an optimal GC content of 40–60%.

### 2.3. SSR Location, GO and KEEG Enrichment

Based on the *N. colorata* genome annotation file (.gff3 Data), the intergenic and gene-intergenic regions were calculated according to the length from the starting to ending position information of gene, CDS and exon regions. The gene ontology (GO) enrichment analysis of CDSs with microsatellites, including GO term mapping, classification and enrichment, was conducted using Blast2GO software [20], WEGO software [21] and the GoSeq program [22], respectively. In order to further study the functions of these genes in terms of the networks of genes and molecules, the CDSs containing microsatellites were aligned to the Kyoto Encyclopedia of Genes and Genomes (KEGG) database (http://www.genom e.jp/kegg, accessed on 10 December 2021) using BLASTx with an e-value of <10^−5^ [23].

### 2.4. Plant Materials and DNA Extraction

A total of nine waterlily cultivars were collected from four localities (originating from Hainan, Guangdong, Xinjiang provinces and Beijing City) (Appendix A). Young leaf materials were immediately preserved with liquid nitrogen, transported to the lab and stored at −80 °C until the DNA was extracted. DNA was then extracted from the samples using a Plant Genomic DNA Kit (Tiangen Biotech, Beijing, China) according to the manufacturer’s instructions. DNA quality and quantity were checked using the Nanodrop 2000 (Thermo Fisher Scientific, Waltham, MA, USA).

### 2.5. Validation of SSR Primer Pairs by PCR Amplification

The 50 primer pairs randomly selected from the 107,152 primers were synthesized by Sangon Biological Engineering Technology & Service Co. (Shanghai, China) for polymorphism validation. SSR-primed polymerase chain reactions (PCRs) were performed in 10 µL reaction volumes, which contained 5uL PCR mixed solution (TransGen Biotec, Beijing, China), 0.3 µL forward primer (10 nmol/L), 0.3 µL reverse primer (10 nmol/L), 3.4 µL double distilled water and 1 µL DNA template. PCR amplification was carried out with the following cycling conditions: an initial denaturation at 95 ℃ for 5 min followed by 30 cycles of denaturation at 94 °C for 30 s, annealing at 50–60 °C (dependent on the different primers) for 25 s and extension at 72 °C for 40 s. The PCR reaction was ended with a 5 min incubation step at 72 °C. The amplified products were separated on 8% polyacrylamide gels with 1 × TBE buffer at a constant voltage of 220 V for 50–60 min and visualized by silver staining.

### 2.6. Data Analysis

After the silver staining of the PCR products, the alleles with the maximum molecular weight were manually recorded in binary format as ‘A’, followed by ‘B’, ‘C’, etc., in decreasing order by molecular weight. The polymorphism information content (PIC) was used to calculate the discriminatory power of primer pairs targeting each SSR locus using Power Marker Software (version 3.25) [24]. Meanwhile, the number of alleles (*Na*), the effective number of alleles (*Ne*), Shannon’s information index (*I*), observed heterozygosity (*Ho*), expected heterozygosity (*He*) and inbreeding coefficient (Wright’s fixation index, *Fis*) were calculated using POPGENE (version 1.3.1) [25]. The phylogenetic relationship among nine waterlily cultivars was constructed in a dendrogram based on similarity coefficients using the program NTSYS-pc (version 2.10) [26]. The clustering map was created based on genetic distances and the unweighted pair group method with arithmetic mean (UPGMA), and the simple matching (SM) coefficient was used to construct a tree.

## 3. Results

### 3.1. SSR Motifs Content in the N. colorata Genome

The *N. colorata* genome was assembled into 1429 contigs (with a contig N50 of 2.1 Mb), and the GC content was 38.59%. The total length of the genome was 409.5 Mb with 804 scaffolds, and it was anchored onto 14 chromosomes. A total of 238,816 SSRs were identified in 14 *N. colorata* chromosomes. Out of these, dinucleotide was the most common type, accounting for 51.67% (n = 123,399), followed by mononucleotide (43.56%, n = 104,031), trinucleotide (3.92%, n = 9354), tetranucleotide (0.65%, n = 1549), pentanucleotide (0.12%, n = 286) and hexanucleotide (0.08%, n = 197) (Figure 1A). The analysis of SSR distribution in 14 chromosomes revealed that (i) the average density was 662.60 SSRs per Mb; (ii) the largest number of SSRs were present on chromosome 1 (n = 30,426, 705.94 SSRs per Mb), followed by chromosome 2 (n = 24,996, 714.17 SSRs per Mb); and (iii) the largest numbers of SSR types were dinucleotide and mononucleotide on chromosome 1 (Figure 1B). In addition, Pearson correlation analysis revealed that chromosome length was significantly positively associated with the number of SSRs in each chromosome (r = 0.982, *p* < 0.01) (Figure 1C).

### 3.2. Characterization of SSR Motifs in the N. colorata Genome

The analysis of the base composition for SSR motifs indicated that (i) A/T were dominant, accounting for 96.76% among the mononucleotide repeats; (ii) AT/TA were the most frequent (43.47%), followed by AG/CT (42.42%), AC/GT (14.04%) and CG/GC (0.08%) among dinucleotide repeats; (iii) AAG/CTT (40.53%) were the most abundant, followed by AAT/ATT (25.34%) and AGG/CCT (12.40%) among the trinucleotide repeats; and (iv) the most abundant repeats among the tetranucleotide, pentanucleotide and hexanucleotide repeats were AAAT/ATTT (42.39%), AAGGG/CCCTT (21.77%) and AACCCC/GGGGTT (71.86%), respectively (Figure 2).

Regarding SSR repeat numbers, repeat times ranging from 5 to 30 times were the most common. Of these, 10 was the most common number of repetitions, accounting for 21.83% (52,145/238,816) of the total SSRs. Interestingly, mononucleotides with 10 repetitions; dinucleotides and hexanucleotides with 6 repetitions; and trinucleotide, tetranucleotide and pentanucleotide with 5 repetitions were dominant among SSR repeat categories. In addition, the SSR number decreased with the increase in repeat times from mononucleotide to hexanucleotide repetitions (Figure 3).

### 3.3. GO and KEGG Enrichment Analysis of CDSs Containing Microsatellite in the N. colorata Genome

A total of 82.20% (196,305/238,816) of SSRs were mainly found in the intergenic regions, followed by 15.76% (37,645/238,816) of SSRs that were found within the genes. Only 2.04% (4866/238,816) in the gene-intergenic regions were distributed in the *N. colorata* genome. The detailed information for this is shown in Appendix A.

Among 2038 CDSs with SSRs, 939 CDSs were annotated and further subjected to a gene ontology (GO) analysis. A total of 793 GO functional terms were assigned, in which 183 GO terms were significantly overrepresented in the sets of genes with SSRs (*p* value < 0.05). In the three categories, protein binding, membranes and the regulation of transcription were the most abundant in the biological process, cellular component and molecular function ontology, respectively (Figure 4A). In addition, with respect to membrane, integral component of membrane, protein binding and regulation of transcription, DNA-templated dominated both in absolute number and rich factor. Meanwhile, the regulation of protein kinase activity and protein kinase regulator activity had the highest rich factor in the top 20 GO terms enrichment analysis results (Appendix A).

A Kyoto Encyclopedia of Genes and Genomes (KEGG) pathway analysis showed that 191 CDSs containing microsatellites were annotated to 72 KEGG pathways, which were further categorized into four major categories (environmental information processing (EIP), genetic information processing (GIP), organismal systems (OS) and metabolism). In brief, pentose and glucuronate interconversions, thermogenesis, the PI3K-Akt signaling pathway and aminoacyl-tRNA biosynthesis were the most abundant in the metabolism, OS, EIP and GIP category, respectively (Figure 4B). In addition, the PI3K-Akt signaling pathway, the mTOR signaling pathway, thermogenesis, the relaxin signaling pathway and pentose and glucuronate interconversions were dominant both in absolute number and rich factor. Meanwhile, the lipoic acid metabolism pathway had the highest rich factor in the top 20 KEGG pathways enrichment analysis results (Appendix A).

### 3.4. Genome-Wide SSR Marker Development

Based on the start positions of the SSR markers, the reference genome physical map, which included 107,152 primer pairs, is described in Appendix A, with results ranging from 540.68 per Mb on chromosome 1 to 198.66 per Mb on chromosome 10, with an average density of 318.67 per Mb (Figure 5). In addition, the distribution of each chromosome showed that these primer pairs were averagely distributed on each chromosome except chromosome 1. Interestingly, there were some regions with a higher density distribution of primers, such as 0–8 Mb, 14–21 Mb, 26–33 Mb and 36–41 Mb on chromosome 1.

### 3.5. Validation Analysis of SSR Primers

Among the 50 primer pairs, 13 primers were practically screened out and used for a polymorphic analysis of nine waterlily species. In total, 30 alleles were found (average 2.308 alleles/primer pair or locus). The *PIC* values of each SSR primer pair ranged from 0.290 to 0.624 (average of 0.475). Among them, Ny-5.2 and Ny-10.1 were the most informative SSR primer pairs, as they had the highest *PIC* values. Meanwhile, Ny-10.1 had the highest value of *Ho*, and Ny-5.2 had the highest values of *Ne*, *I* and *He*. The detailed information for this is shown in Table 1.

### 3.6. Cluster Analysis

The phylogenetic tree indicated nine waterlily cultivars that were divided into two clusters; Cluster I and Cluster II. *Nymphaea ‘Black Beauty’*, *N*. *colorata* and *Nymphaea lotus*, traditionally classified as tropical waterlilies, were clustered into two subgenera (*Brachyceras* and *Lotos*) compared to *Nymphaea ‘Islamorada’* and *Nymphaea ‘Tina’*. In addition, *Nymphaea ‘Princess Elizabeth’*, *Nymphaea candida*, *Nymphaea ‘Colorado’* and *Nymphaea Mexicana*, traditionally classified as hardy waterlilies, were divided into two clades (Figure 6).

## 4. Discussion

Prior to this study, due to the lack of genomic information on waterlily, the mining and development of molecular markers were limited, and only a limited number of molecular markers have been used in waterlily [13,27,28]. In this study, we were the first to scan the genome-wide SSRs of *N. colorata*. A total of 238,816 SSRs were identified, presenting at a density of 583.75 SSRs per Mb. This density was extremely low compared with those reported in previous studies on the dicotyledon species *Prunus mume* (234.03 Mb, 794 SSRs/Mb) [29] and pomegranate (308.438 Mb, 1,230.6 SSRs/Mb) [30] but high in comparison to those of *N*. *nucifera* (804.648 Mb, 236.41 SSRs/Mb) [6] and *Malus domestica* (703.358 Mb, 40.8 SSRs/Mb) [31]. Our study supports the idea that SSR density is negatively correlated with genome size in plants [32].

Moreover, these motif types and their proportions in the *N. colorata* genome are in close agreement with the patterns observed in aquatic plants, such as Asian lotus (*N. nucifera*) and American lotus (*Nelumbo lutea*) [33], in which di- and tri-nucleotide repeat motifs were the most abundant. Moreover, the AT/TA and A/T are the most abundant types among the dinucleotide and mononucleotide repeat motifs, accounting for 43.47% and 96.76%, respectively. This result agrees with that of the dicot species (e.g., pomegranate, peanut and cucumber) [34,35,36]. In addition, the SSR number decreased with the increase in repeat times from mononucleotide to hexanucleotide repetitions based on a genome-wide scan of microsatellites in *N. colorata*. This finding presents similar characteristics to peanut (*Arachis hypogaea*) and asparagus (*Asparagus officinalis*) [35,37].

In previous studies, the distribution of SSRs varied in different regions across a genome and had a higher degree of abundance in noncoding than in coding regions in various animals and plants [38,39,40]. A similar feature was also observed in *N. colorata*. In our studies, 82.20% (196,305/238,816) of the SSRs were distributed in intergenic regions, and only 1.02% (2442/238,816) of SSRs were abundant in exon and CDS regions. Furthermore, we calculated that SSR repeat types ranged from mononucleotide to hexanucleotide. Interestingly, trinucleotides were much more abundant than other nucleotides and accounted for 63.84% (1559/2442) (Appendix A). This finding conveyed to us that trinucleotide SSRs in CDS regions were significantly abundant in *N. colorata*, which did not cause frameshifts and were not notably influenced by coding status [41]. This phenomenon is similar to king cobra (*Ophiophagus hannah*) [42].

A GO analysis revealed that CDSs containing SSRs were mainly associated with signal identification, such as protein kinase activity (e.g., GO: 0045859, the regulation of protein kinase activity; GO: 0019887, protein kinase regulator activity), which might play a major role in response to a variety of stimuli such as phytohormone treatment and temperature stress [43]. A KEGG pathway analysis showed that CDSs within SSRs play a major role in metabolism (ko00785, lipoic acid metabolism), which might strengthen the antioxidant network of the cells [44]. In the future, it will be necessary to explore SSR diversity in CDS regions when comparing the special function (e.g., freezing resistance and disease resistance) of different waterlily species originating from different geographical habitats.

In previous studies, SSR markers were effectively applied to the diversity of local and wild lotus varieties [45,46,47]. To analyze the genetic relationships between nine waterlily cultivars, a total of 13 pairs of primers targeting polymorphic SSR markers were screened out. In total, 30 alleles (average 2.308 alleles/primer pair or locus) were found. The UPGMA dendrograms show that the nine waterlily cultivars were divided into two clusters, which is consistent with the eco-physiological classification (tropical waterlily and hardy waterlily) [3]. This finding indicated that 13 pairs of primers targeting polymorphic SSRs are useful for identifying different waterlily species due to the transferability of SSR markers. Many studies have demonstrated the utility of the transferability of SSRs for the analysis of intra- and inter-specific genetic diversity and species identification [48,49,50]. These SSR markers in waterlily might also be applicable to cross-genera genotyping or to genotyping in other closely related plant species. In addition, the values for *PIC*, *Ho*, *He*, *Ne* and *I* indicated that Ny-5.2 and Ny-10.1 are highly polymorphic and could potentially be used in genetic diversity. In the future, the most effective SSR primers could be developed as more genomes of waterlily species are sequenced.

## 5. Conclusions

Here, we report the first comprehensive study of SSR density, occurrence frequency and GO and KEGG enrichment analysis based on the *N. colorata* genome. A total of 238,816 SSRs were identified in 14 *N. colorata* chromosomes with an average density of 662.60 SSRs per Mb. The dinucleotide was the most common type, and AT-rich repeats prevail in the *N. colorata* genome. In GO and KEGG enrichment analysis of CDSs containing microsatellites, signal recognition (e.g., GO: 0045859 and GO: 0019887) and metabolism (ko00785, lipoic acid metabolism) were significantly enriched, respectively. In addition, the large amount of SSR marks (n = 107,152) enriches molecular markers in waterlily. Among these, the 13 novel candidate SSR markers used in this study will be useful for genetic diversity and phylogenetic analysis to differentiate between waterlily species, hybrids and even lineages.

## Figures and Tables

**Figure 1 genes-13-01782-f001:**
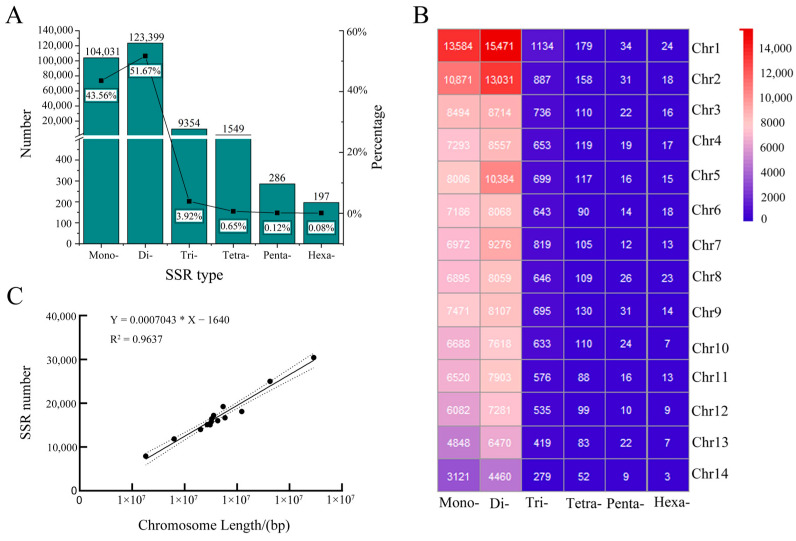
Number of SSRs identified on each chromosome in the *N. colorata* genome. Number and percentage of SSR markers ranging from mononucleotide to hexanucleotide repeats in the whole genome (**A**). SSR numbers ranging from mononucleotide to hexanucleotide repeats in each chromosome (**B**). Pearson correlation analysis between SSR number and each chromosome length (**C**).

**Figure 2 genes-13-01782-f002:**
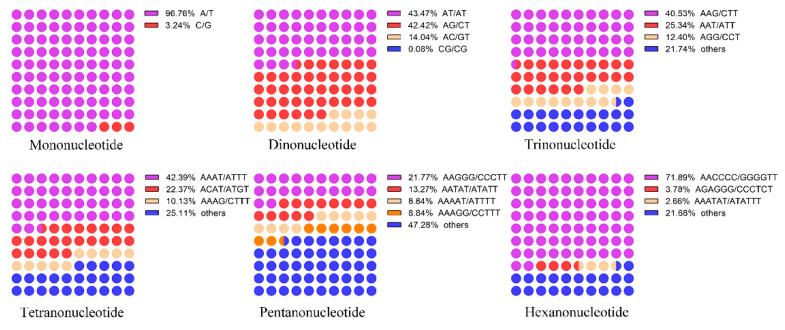
The percentage of mononucleotide to hexanucleotide SSR motifs in the *N. colorata* genome.

**Figure 3 genes-13-01782-f003:**
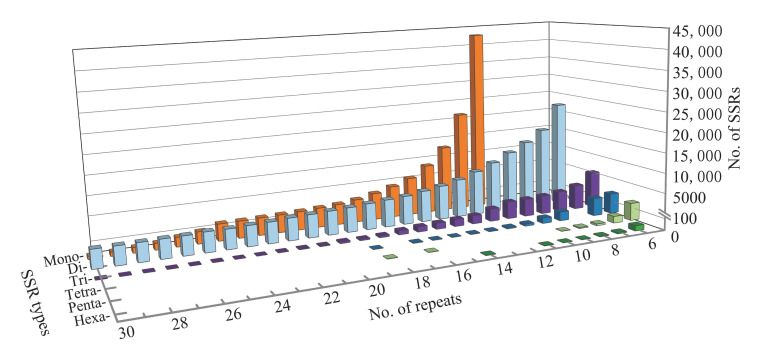
The number of repeat types with respect to the number of repeat motifs of SSRs in the *N. colorata* genome.

**Figure 4 genes-13-01782-f004:**
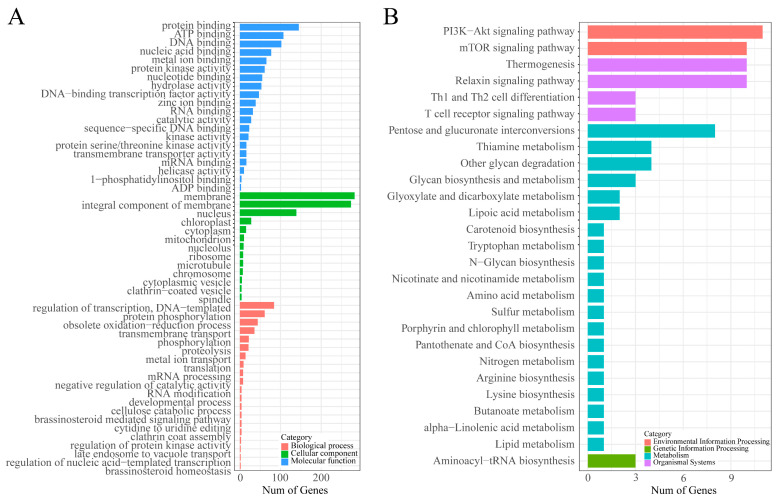
GO and KEGG enrichment analysis. GO classifications of CDSs with SSRs (**A**). KEGG pathway classifications of CDSs with SSRs (**B**).

**Figure 5 genes-13-01782-f005:**
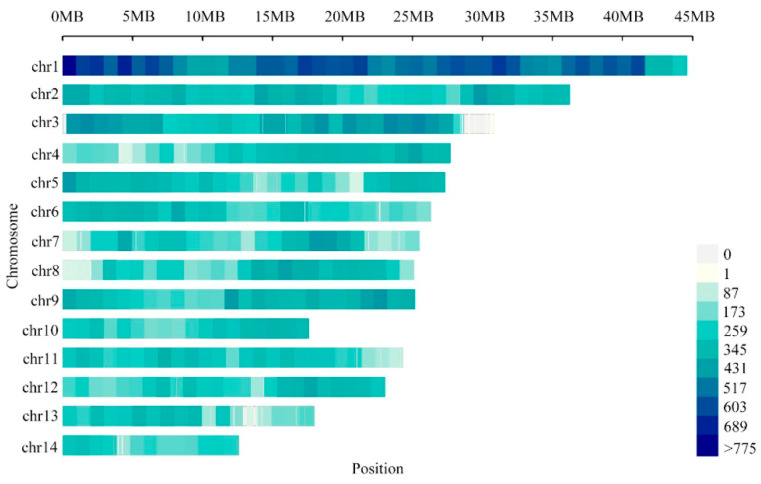
Overview of the high-density SSR primer pairs physical map in *N. colorata*. The bar represents the number of SSR primer pairs within a 1 Mb window.

**Figure 6 genes-13-01782-f006:**
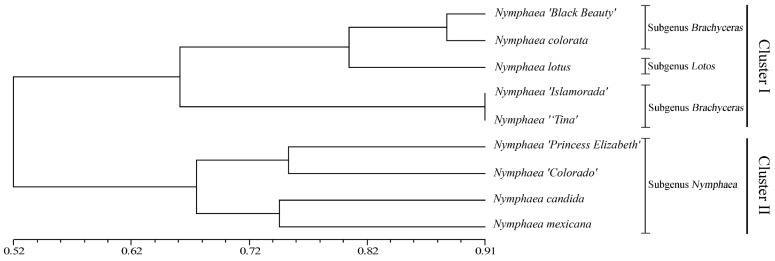
Cluster dendrogram of SSR markers of nine waterlily cultivars.

**Table 1 genes-13-01782-t001:** Characteristics of the 13 polymorphic microsatellite loci and primer sets.

Locus	Core Motif	Chr.	Primer Sequences (5′–3′)	*Na*	*Ne*	*I*	*Ho*	*He*	*Fis*	*PIC*
Ny-2.1	(T)13	2	F: AGAGCTGAGATTGGTTTGAAGC R: TCAGCGATTTCTTCTTGGGAT	2.000	1.800	0.637	0.000	0.471	1.000	0.444
Ny-2.2	(CT)10	2	F: TTTGTGGTGCGACTTCTTGC R: GTTAACAGCAGCCTTCACCG	3.000	1.409	0.557	0.333	0.307	−0.149	0.290
Ny-3.2	(A)12	3	F: CTATGGACAACACATGCCGC R: CACGAGCAACAAGACCAGTA	2.000	1.800	0.637	0.000	0.471	1.000	0.444
Ny-4.2	(GA)33	4	F: CACGGCGAGGGGACAATATA R: CCGCCAATTCCACCATTCAT	3.000	1.906	0.787	0.333	0.503	0.299	0.475
Ny-5.1	(A)13	5	F: AAAGAACTATGCGGACCTGC R: CAAGCTCAGGACATGGTTCG	2.000	2.000	0.693	0.111	0.529	0.778	0.500
Ny-5.2	(TC)20(TA)25	5	F: CCGCAGTTAGTGTCACATGG R: CGCGTTCTCCTTTGCCAATA	3.000	2.656	1.037	0.222	0.660	0.644	0.624
Ny-6.2	(T)10…(AT)10	6	F: AATCAATGCTTCCATGGCCG R: TCATGTCCGGGATTCTAGGC	2.000	1.976	0.687	0.000	0.523	1.000	0.494
Ny-9.1	(TA)17(GA)14	9	F: ACCAAGGACTGCGAGTGTAT R: ATTTGAGTTGAGGGTTGCCG	2.000	1.670	0.591	0.556	0.425	−0.385	0.401
Ny-10.1	(AG)8… (AG)10	10	F: CCCAGCATCGTAAATGACCG R: TTGGAGGAGGAGGAGATTGC	3.000	2.418	0.981	0.667	0.621	−0.137	0.586
Ny-11.1	(AG)6…(T)11	11	F: AGCGTCACAACACTCCACTA R: AGGATTAGATGGGGCTCTGC	2.000	1.906	0.668	0.333	0.503	0.299	0.475
Ny-12.1	(TC)10(TA)13	12	F: AGGAGAAAACAGAGTGGGGC R: AGCATGCATGTATTCCCCAT	2.000	1.800	0.637	0.222	0.471	0.500	0.444
Ny-13.1	(CT)6(CA)9	13	F: CAGATGCAAGGATGGGAAGC R: GCAATGGGGATGATGAAGGC	2.000	2.000	0.693	0.556	0.529	−0.111	0.500
Ny-13.2	(TC)20(TA)12	13	F: GCCTACCCATGTCCTCTGAT R: CCCTGTTCTGTTTGTGTTGC	2.000	1.976	0.687	0.000	0.523	1.000	0.494
Mean	-	-	-	2.308	1.947	0.715	0.256	0.503	0.441	0.475

## Data Availability

All data presented in this study are available in the article itself or in the provided Appendix A files.

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
