# Peer review of "Genome-Wide Survey and Analysis of Microsatellites in Waterlily, and Potential for Polymorphic Marker Development"

_genes, 2022, doi:10.3390/genes13101782_

Round 1

Reviewer 1 Report

This is an interesting study. The authors have identified and characterized simple sequence repeats (SSRs) from the draft genome of a typical tropical waterlily species, Nymphaea colorata, and developed highly polymorphic SSR markers. The results are clearly presented, and the manuscript reads well; however, there are some issues that the authors need to address. Please see my comments:

1- In Materials and Methods, section 2.1, most of the information in this section is a characteristic of results. Please rewrite this section.

2- In Materials and Methods, section 2.2, The authors should mention how they identified the intergenic and gene-intergenic regions and the type of annotation files.

3. Please check the tools used in this study and add the tool version.

4- Line 130, please remove the text "The text continues here.".

5- In figure 1A, please replace the x-axis title "Nucleotide category" with "SSR type".

6- I suggest improving the quality of the figures to increase the readability.

7- Most references are old, and recent five-year references should be included.

Reviewer 2 Report

The authors represented mining of microsatellites of 1-6 bp core motifs from the waterlily genome, in silico characterization of the microsatellites, and the discovery of polymorphic microsatellites proved by SSR marker assays. According to the Introduction, these are novel with biological significance in terms of ornamental and complement the limited availability of molecular markers in waterlily.

The manuscript contents largely follow the form of studies on microsatellites, which has been established in the research community while Discussion contents are relatively poorly written than other sections. Results/Discussion sections lack any description about specific findings on the mined microsatellites on particular specific genes related to the important traits of waterlily (it might be homeotic genes, genes for pigment biosynthesis genes, or genes for aquatic plant physiology). As the waterlily genome is available, the re-sequencing strategy (mainly uses whole genome SNPs) enables to be taken in diversity and genetic analyses. Hence, the authors should more emphasize the advantages of microsatellite markers based on its genome-wide mining.

Here are points to be addressed technically in text

(1)    English needs extensive corrections by a native-level biologist

(2)    The title is grammatically correct? “Mining of whole genome microsatellites in waterlily and its application to polymorphic marker development” or something like that?

(3)    Introduction needs a stronger logical story that definitely shows the necessity of genome-wide microsatellite mining in waterlily.

(4)    Waterlily cultivars look belonging to multiple species (Nymphaea XXXX). Then all of these are not the identical species? Then the polymorphisms discovered comprise intraspecific size polymorphisms and interspecific size polymorphisms or not? If it contains intraspecific size polymorphisms, this point needs to be described technically.

(5)    There are many whole genome microsatellite studies in plant species. What about the novelty in aquatic plants and in closely related plant taxonomy? Including the basic characteristics of microsatellites such as the frequencies and core motif compositions should be discussed more.

(6)    Microsatellite mining results are dependent on the method/tool applied. The authors used MISA, but what about the perfectness/imperfectness of the mined microsatellites? Including this information, the authors had better discuss the technical limitation in this study. Preferably, it is better to show a reference result for the Arabidopsis genome or something else in the same computational parameters.

(7)    The format of reference list is completely correct?

Round 2

Reviewer 2 Report

The authors largely addressed the issues, and the manuscript is in a better shape. The reviewer leaves comments on the revised manuscript. Hope that these constructive criticisms are useful for improving this paper before acceptance.

(1)    The entire text in English needs further corrections/proofreading to provide better readability and strict meanings of expressions. For showing an example, the reviewer attached an English-proofreading for the Abstract (only). There were errors in choosing technical terms and the misuse of “,”. There was a few minor information of results that are not needed to describe in Abstract. The best option might be to receive a professional English proofreading service throughout the text.

(2)    The genotyping by the SSR markers were conducted across the nine Nymphaea species. These could have been done due to the transferability of SSR markers. The authors represented cross-species transferability experimentally. However, there is a potential that the SSR markers in waterlily might be applicable to cross-genera genotyping or to genotyping in other closely related plant species. This point can be described in Introduction or Discussion to further increase the scientific value of this study. Cross-genera/cross-species transferability of plant SSR markers were documented in the following papers.

Manoj K Rai et al. (2013) 40: 5067-71. doi: 10.1007/s11033-013-2608-1

Endo et al. Euphytica (2017) 213:56. doi: 10.1007/s10681-017-1846-z

     The transferability of SSR markers is useful for species identification. A paper (Tuler et al. 2015) described an example in plant species.
                   Tuler et al. Mol Biol Rep (2015) 42: 1501-13. doi: 10.1007/s11033-015-3927-1
